# 3D Imaging and Quantitative Characterization of Mouse Capillary Coronary Network Architecture

**DOI:** 10.3390/biology10040306

**Published:** 2021-04-07

**Authors:** Nabil Nicolas, Etienne Roux

**Affiliations:** Univ. Bordeaux, Inserm, UMR1034, Biology of Cardiovascular Diseases, F-33600 Pessac, France; etienne.roux@u-bordeaux.fr

**Keywords:** coronary, capillary network, optical clearing, light sheet microscopy, 3D imaging, 3D image processing

## Abstract

**Simple Summary:**

The cardiovascular system is composed of two physically and functionally linked parts, the heart and the vascular network, composed of the arteries, capillaries, and veins. In this system, the heart maintains the blood flow, including its own through the coronary vascular network, bringing oxygen and nutrients required for its activity. Microvascularization constitutes the distal part of this network and is composed of small vessels, i.e., arterioles and capillaries. Coronary microvascular diseases are associated with the development of cardiac dysfunction. Identification of the coronary microvascular network structure is hence of critical importance for the understanding of heart diseases. The aim of our study was to establish an accessible methodology for 3D imaging and 3D processing to quantitatively characterize the capillary coronary network architecture in mice, a species largely used as an animal model of cardiac failure. This study proposes a standardized methodology for 3D image processing that allows the quantification of the 3D capillary architecture. Based on an open source software, the whole process is easily accessible for biologists. It is hence useful for the study of cardiovascular diseases and the development of future therapies.

**Abstract:**

Characterization of the cardiac capillary network structure is of critical importance to understand the normal coronary functional properties and coronary microvascular diseases. The aim of our study was to establish an accessible methodology for 3D imaging and 3D processing to quantitatively characterize the capillary coronary network architecture in mice. Experiments were done on C57BL/6J mice. 3D imaging was performed by light sheet microscopy and confocal microscopy on iDISCO+ optical cleared hearts after labelling of the capillary endothelium by lectin injection. 3D images were processed with the open source software ImageJ. Non-visual image segmentation was based of the frequency distribution of the voxel greyscale values, followed by skeletonization and distance mapping. Capillary networks in left and right ventricles and septum were characterized by the volume network density, the fractal dimension, the number of segments and nodes and their ratio, the total network length, and the average length, diameter, and tortuosity of the segments. Scale-dependent parameter values can be impacted by the resolution limit of the 3D imaging technique. The proposed standardized methodology for 3D image processing is easily accessible for a biologist in terms of investment and difficulty level, and allows the quantification of the 3D capillary architecture and its statistical comparison in different conditions.

## 1. Introduction

Oxygen and nutrient delivery to the heart is ensured by blood flow through the coronary vascular network, composed by arteries, capillaries, and veins. The structure of this network is a major determinant of heart metabolism and functional capacities. The microvascularization constitutes the distal part of this network and is composed of the arterioles and capillaries with a diameter lower than 100-150 µm. Exchanges between blood and cardiac tissue take place in the capillary network. Structural or functional alterations of the coronary microvascularization are referred as coronary microvascular diseases (CMDs) [1]. These CMDs are associated with the development of cardiac pathologies, including angina and heart failure [2].

Identification of the coronary microvascular network structure is hence of critical importance for the understanding of normal coronary functional properties and the understanding of several cardiac dysfunctions. Historically, most of the studies that intended to analyze human and animal coronary capillary networks have been done in 2D. In humans, the coronary microcirculation is hardly detectable by standard imaging techniques [1]. 2D angiography has been used to measure the large coronary vessel diameters, but not microvessels [3]. In mice, histological techniques have been used for 2D measurements of capillary diameter and density [4,5]. During the last few years, the use of 3D imaging techniques to study blood vessels has increased. For example, such techniques have been used to study and quantify the renal arterial and arteriolar system [6] and the brain vascular system [7,8]. However, to our knowledge, no studies have been published that have quantified the coronary capillary 3D architecture. The aim of our study was to establish an accessible methodology for 3D imaging and 3D processing to quantitatively characterize the capillary coronary network architecture in mice, a species largely used as an animal model of cardiac failure [9].

For 3D imaging, our strategy was to use 3D photonic imaging to visualize the coronary capillary system on intact mice hearts. A major problem is light scattering that limits light penetration in the tissue. Clearing methods that homogenize the refractive index have been developed in the past decade to overcome this difficulty [10]. Among the several techniques available for optical clearing, we chose the iDISCO+ method [11]. Its main advantages are easy access, short clearing time and high clearing capacity. Labelling of the capillary network was ensured by lectin coupled to a fluorophore, as lectin has been shown to be a specific marker of the capillary endothelium [12]. Cleared samples were imaged by light sheet microscopy (LSM) that allows large 3D visualization in intact samples [10]. Since confocal microscopy can also be used on cleared tissues, we also performed complementary analysis with this imaging technique.

For 3D image processing, we developed a standardized methodology to extract 3D quantitative data with the open source software ImageJ. Capillary network structure was characterized with nine parameters divided in two types: (1) global parameters related to the overall structure of the vascular system (vascular density, fractal dimension, number of segments, total length, number and percentage of node), and (2) architectural parameters related to the structure of each segment of the vascular system (length, diameter, and tortuosity). Comparisons between the left (LV) and right (RV) ventricles and the septum (S) were made, as no comparative information about mouse physiological coronary capillaries regarding the different parts of the heart was found in the literature. The only comparison found concerned the rat LV and feline LV and RV [13,14].

## 2. Materials and Methods

### 2.1. Animal Model

Experiments were performed on 8-week-old C57BL/6J mice (*n* = 7). Food and water were available ad libitum, with a 12 h dark/light cycle.

### 2.2. Arterial Pressure Measurements

Systolic (SAP), diastolic (DAP), and mean arterial pressure (MAP) were measured on vigil animals using the CODA^®^ High Throughput System (Kent Scientific corporation, CT, USA). This non-invasive method is based on the analysis of the caudal blood flow by a sensor cuff after the complete occlusion of the blood flow and its progressive recovery controlled by an occlusion cuff placed at the basis of the tail. Each mouse was placed in an animal holder on a warming support in order to vasodilate the caudal vasculature. The tail external temperature was monitored and the recording began when it reached 32 °C. Measurements were realized on 15 consecutive cycles over 15 min, and the mean SAP, DAP, and MAP were calculated on these 15 cycles.

### 2.3. Mice Preparation

Each mouse received a retroorbital intravenous 100 µL injection of lectin coupled to a fluorophore (*Lycopersicon esculentum Lectin* DyLight649^®^ 1 mg/mL, Vector Labs, Eurobio Scientific, Les Ulis, France) with an insulin syringe. Ten minutes after the injection, an intraperitoneal 100 µL injection of isosorbide dinitrate (Risordan^®^ 10 mg/10mL, Medisol, Lyon, France) was made with 25G needle to dilate the vessels. The mouse was then euthanized by an intraperitoneal injection of 300 µL sodium pentobarbital (Exagon^®^ à 400 mg/mL, Axience, Pantin, France) diluted in physiological saline solution. After death, a sternotomy was carried out to catheterize the left ventricle. A perfusion of physiological solution at 80 mm Hg pressure for 3 min was done to remove the blood from the vasculature. A second perfusion of 4% formalin (10% neutral buffered formalin, DiaPath, Martinengo, Italy) was done to fix the tissues. The heart was delicately removed and placed in paraformalin overnight at 4 °C. A negative control was done in similar conditions, but without lectin injection.

### 2.4. Optical Clearing

Optical clearing of the tissues was performed using the iDISCO+ method [11]. The technique consists of a methanol pre-treatment, followed by methanol and dichloromethane permeabilization and lipid removal, and RI matching in dibenzyl ether.

Each heart was pre-treated by successive immersions in 20%, 40%, 60%, and 80% methanol solutions for 1 h each at room temperature, then left overnight in a pure methanol solution (Methanol ≥ 99.5%, GPR RECTAPUR^®^, VWR Chemicals, Fontenay-sous-Bois, France) at room temperature. Permeabilization and lipid removal were then performed in a 2/3 dichloromethane (Dichloromethane, anhydro ≥ 99.8%, with 40–150 ppm amylene, Sigma-Aldrich, Lyon, France) and 1/3 methanol solution for 3 h at room temperature. The remaining methanol was removed by two 15 min baths in pure dichloromethane solutions at room temperature. Last, RI matching was done by leaving the sample in a dibenzyl ether solution (Benzyl ether 98%, Sigma-Aldrich) for a few hours at room temperature, then kept at 4 °C until imaging. 

### 2.5. Shrinkage Measurement

The iDISCO+ technique has been shown to generate a shrinkage of the tissues [10]. On each sample, the shrinkage was estimated from two orthogonal 2D-images taken before and after clearing. Length (l), width (w), and thickness (t) were measured using ImageJ/Fiji software. These measurements were used to calculate an estimated heart volume, considered as an ellipsoid, thanks to the following equation:$$Cardiac\;Volume=\frac{4}{3}\times \pi \times \frac{l}{2}\times \frac{w}{2}\times \frac{t}{2}$$

The shrinkage percentage was calculated for the length, the width, the thickness, and the ellipsoidal volume as the ratio of each parameter value after and before optical clearing on the parameter value before optical clearing.

### 2.6. Image Acquisition

#### 2.6.1. Light Sheet Microscopy 

Each heart was mounted in ethyl cinnamate solution in a specific device adapted to the light sheet microscope. The ultramicroscopy was done using the system from LaVision BioTec (Bielefeld, Germany) equipped with a 639 nm (70 mW) laser line, a sCMOS Andor camera, and a 0.5 NA 2× objective with a deeping lens, with 6.3 zoom. The tissue was illuminated laterally by three horizontal sheets. Exposition time was 200 ms. Emitted fluorescence was collected at 690 nm. At the end of the acquisition, the stack was automatically reconstructed in 16 bits format. Capillary network imaging was made by sampling three 1080 × 1280 × 300 µm^3^ parallelepipedic sections in the left ventricle (LV), the septum (S), and the right ventricle (RV) of the heart. The system spatial resolution was 1 µm (x, y) and 4 µm (z). Step size used was 2 µm. Voxel dimensions were 0.5 µm (x, y) and 2 µm (z).

#### 2.6.2. Confocal Microscopy 

3D imaging of cleared tissues is usually done either by light sheet microscopy or confocal microscopy [10]. To compare the potential variations due to the imaging technique, LV confocal microscopy 3D imaging was performed on one heart sample used for light sheet microscope imaging. A thin section of LV was cut and mounted in a specific device adapted to confocal microscopy for cleared samples. The tissue was illuminated by a laser at 639 nm and the emitted fluorescence at 690 nm was collected by a HCX PL APO CS 20.0 × 0.70 IMM UV objective. At the end of the acquisition, the stack was automatically reconstructed in 16-bits format. Capillary network imaging was made by sampling three 387 × 387 × 105 µm^3^ parallelepipedic sections. The system spatial resolution was 279 nm (x, y) and 1284 nm (z). The step size used was 630 nm. Voxel dimensions were 380 nm (x, y) and 630 nm (z). 

### 2.7. Image Processing

Commands and functions for each step of the image processing are given in Appendix A.

The stacks were converted to 8 bits format (256 grey levels) and image processing was performed using the open-source ImageJ/Fiji software (ImageJ 2.1.0/1.53h/Java 1.8.0_66 (64 bit)) and several of its plugins.

#### 2.7.1. Segmentations

The extraction of the two structures of interest, i.e., the whole cardiac tissue and the capillary network, was performed by a segmentation technique based on two greyscale-defined thresholds. For each sample, the threshold values were determined from the frequency distribution of the voxel greyscale values. This distribution showed the existence of three voxel populations corresponding, from 0 to 255 grey values, to (1) the image background, (2) the non-labelled cardiac tissue, and (3) the lectin-labelled capillary network, respectively. The first threshold, segmenting the cardiac tissue from the background, was set at the inflexion point of the distribution curve between the first and second population. Due to the overlap of the 2 voxel populations corresponding to the cardiac tissue and the lectin-labelled capillary network, no inflexion point was available for segmenting the capillary network, making difficult to identify an objective threshold value. However, the negative control showed that the non-labelled cardiac tissue is an homogenous population of voxels following a Gaussian distribution. Hence, on each sample, a Gaussian fit of the voxel frequency distribution was performed to determine the mean and standard deviation (SD) of the voxel population corresponding to the non-labelled cardiac tissue, and the threshold value was set at mean + 1.96 SD. Due to the mathematical properties of the normal distribution, such a threshold value ensured the exclusion of 97.5% of the voxels that do not correspond to the capillary network. Frequency distribution curves of the negative control and of one representative lectin-labelled sample are given in Figure 1a–d.

#### 2.7.2. Filtering 

After binarization of the segmented images, a 3D median filter set at 3 voxels for (x, y, z) was applied to suppress artefactual isolated voxels.

#### 2.7.3. Skeletonization and Distance Mapping 

After segmentation, a skeletonization of the segmented capillary network image was performed using the *BoneJ* plugin [15]. In the resulting skeleton, each segment of the network was transformed into a one voxel line corresponding to the center of the segment and identified by its own orthonormal coordinates (x, y, z). In parallel, distance mapping was performed on the segmented image, using the *3D Suites* plugin [16]. Basically, distance mapping computes for each voxel of the segment axis the radius of the segment at that point. Skeleton and distance map images were fused before skeleton analysis. This provided information on the number of segments and the number of junctions between segments and, for each individual segment, the orthonormal coordinates of its extremities, its length, its mean radius, and its Euclidian distance, i.e., the linear distance between the two ends of the segment. Segments with a Euclidean distance equal to 0, are considered as artefactual loops as their starting and end points are the same, and segments with a radius inferior to 1 voxel, reaching the resolution limit, were removed from the data. The mean percentage of these artefactual loops was 4.62% of the total number of segments. Representative images of LV capillary network of image processing are shown in Figure 1e,f.

### 2.8. Data Analysis 

The data obtained from skeletonization and distance mapping were used to determine 6 “global” parameters, i.e., considering the capillary network as a whole (vascular density, fractal dimension, number of segments, number and percentage of nodes, and total length), and 3 “topological” parameters, i.e., calculated from each segment of the network (length, diameter and tortuosity).

#### 2.8.1. Cardiac Volumes and Vascular Density

The total cardiac volume (non-labelled cardiac tissue + lectin-labelled capillary network volumes) and the capillary network volume were obtained from their respective segmented binary images. For each object, the volume, in mm^3^, was calculated by multiplying the number of white voxels, corresponding to the volume object in voxels, by the volume of one voxel in mm^3^.

The volume vascular density, expressed as %, was calculated according to the following equation:$$Vascular\;density=\frac{Capillary\;volume}{Total\;volume}$$

#### 2.8.2. Fractal Dimension

The fractal dimension measures the ability of a self-similar structure to fill the three-dimensional space it occupies. The fractal dimension of the capillary network was calculated on the skeletonized image using the Minkowski–Bouligan method [17], also called Box Counting method, with BoneJ plugin.

#### 2.8.3. Normalized Number of Segments 

The number of segments obtained from the skeletonization was normalized to the total cardiac tissue volume (in mm^3^).

#### 2.8.4. Normalized Total Capillary Length

The length of each segment of the skeleton was obtained in µm from the skeletonization, and the normalized total length of the network, expressed in m.mm^-3^, was calculated by summing up all segment lengths converted in meter, normalized to the total cardiac tissue volume in mm^3^.

#### 2.8.5. Normalized Number and Percentage of Nodes

The normalized number of nodes was defined as the number of junctions formed by the capillary network normalized to the total cardiac tissue volume (mm^3^). The percentage of nodes was calculated by dividing the number of nodes by the number of segments. The node/segments ratio depends on the number of segments connected at each node, and can hence be used as an index of the connectivity of the network.

#### 2.8.6. Segment Diameter

The segment mean radii were obtained in µm from the skeletonization in µm, and the segment mean diameters were calculated by multiplying the mean radii by 2. 

#### 2.8.7. Tortuosity

The tortuosity, used as an index representing the twisting of each capillary segment, was calculated using the following equation:$$Tortuosity=\frac{Segment\;lenght}{Euclidian\;distance}$$

### 2.9. Statistical Analysis

Statistical analyses were done using GraphPad Prism^®^ software (Prism 9.0.1 version), (San Diego, CA, USA).

#### 2.9.1. Global Parameters 

Vascular density, fractal dimension, number of segments, number and percentage of nodes, and total length are expressed as mean ± standard deviation (SD). Statistical comparisons between LV, S and RV were done with Kruskal–Wallis non-parametric tests followed, if relevant, by post-hoc Dunn’s test. Results were considered as statistically significant for *p* < 0.05.

#### 2.9.2. Topological Parameters

For each topological parameter, i.e., segment length, diameter and tortuosity, frequency classes were defined and the relative frequency distribution was calculated. The frequency distributions were adjusted by non-linear regression. For each parameter, the best fitting curve of the frequency distribution was determined by statistical comparison of Gaussian versus one-phase exponential decay equations with *F* test using GraphPad Prism. Statistical comparisons between LV, S and RV were performed with *F* test. Results were considered as statistically significant for *p* < 0.05.

## 3. Results

### 3.1. Cardiovascular Parameters

Arterial pressures (SAP, DAP, MAP) and cardiac rhythm were measured on six mice. Mean ± SD are given in Table 1.

### 3.2. Shrinkage

The shrinkage percentage in length, width, thickness and ellipsoidal volume was calculated on seven hearts, and are given in Table 2. Shrinkage values were very similar in length, width, and thickness, and their statistical comparison by Kruskal–Wallis test showed no significant difference, indicating that the shrinkage was isotropic.

### 3.3. Global Parameters

#### 3.3.1. Vascular Density

Mean vascular density was calculated for LV, S, and RV on 6 mice. Results are presented in Figure 2a and mean ± SD are given in Table 3. The vascular density was significantly higher in the LV compared to S, but not between LV and RV, nor between RV and S.

#### 3.3.2. Fractal Dimension

The fractal dimension was calculated on the skeletonized images of the LV, the S, and the RV of six hearts. According to the Minkowski–Bouligan method, the box counting results were fitted by a simple linear regression (R² = 0.99), which allowed us to calculate the fractal dimension as the curve slope value. Results are presented in Figure 2b and mean ± SD are given in Table 3. Statistical comparison showed no significant difference between LV, S, and RV.

#### 3.3.3. Normalized Number of Segments 

The number of segments normalized to the volume of cardiac tissue was calculated for the LV, the S, and the RV of six hearts. Results are presented in Figure 2c and mean ± SD are given in Table 3. Statistical comparison showed no significant difference between LV, S, and RV.

#### 3.3.4. Total Capillary Length

The total capillary length was calculated for the LV, the S, and the RV of six hearts. Results are presented in Figure 2d and mean ± SD are given in Table 3. Statistical comparison showed no significant difference between LV, S, and RV.

#### 3.3.5. Number and Percentage of Nodes

The number and percentage of nodes were calculated for the LV, the S, and the RV of six hearts. Results are presented in Figure 2e,f, and mean ± SD are given in Table 3. For the normalized number of segments, no significant difference was found for the normalized number of nodes between LV, S and RV. However, the percentage on nodes was significantly higher in RV compared to S, but not between LV and RV, nor between LV and S.

### 3.4. Topological Parameters

#### 3.4.1. Segment Length

Segment length distributions in LV, S and RV (*n* = 6) were fitted by the following one-phase exponential decay equation: $$F=\left(F_{0}-F_{\infty }\right)\times e^{\left(-K\;\times \;L\right)}+F_{\infty }$$
where *L* is the segment length (in µm), *F* is the relative frequency (in %), $F_{0}$ is *F* for *L* = 0, $F_{\infty }\;$is *F* for *L* = ∞, and *K* is the rate constant (in µm^−1^). From the rate constant *K* the length constant *λ* was defined as 1/*K*, in µm, and used as an index of length distribution. The shorter *λ* is, the higher the proportion of sort segments is. Curves are presented in Figure 3a and *λ* values are given in Table 4. The median values are 12.34, 12.88, and 12.01 for LV, RV, and S, respectively. Statistical comparisons showed no significant differences for *λ* between RV, S, and LV.

#### 3.4.2. Diameter

Diameter length distributions in LV, S, and RV (*n* = 6) were fitted by the following Gaussian equation: $$F=A\times e^{\left(-0.5\times {\left(\left(d-µ\right)/\sigma \right)}^{2}\right)}$$
where *d* is the segment diameter (in µm), *F* is the relative frequency (in %), *A* is the maximal frequency, µ is the diameter mean value (in µm), and *σ* is the standard deviation (in µm). Curves are given in Figure 3b and µ and *σ* are given in Table 4. Statistical comparisons showed that µ was significantly lower in S versus RV and LV, and lower is LV versus RV.

#### 3.4.3. Tortuosity

Tortuosity distributions in LV, S and RV (*n* = 6) were fitted by the following one-phase exponential decay equation: $$F=\left(F_{0}-F_{\infty }\right)\times e^{\left(-K\;\times \;t\right)}+F_{\infty }$$
where *t* is the segment tortuosity (unitless), *F* is the relative frequency (in %), $F_{0}$ is *F* for *t* = 0, $F_{\infty }\;$is *F* for *t* = ∞, and *K* is the rate constant (in µm^1^). From the rate constant *K* the tortuosity constant *τ* was defined as 1/*K*, and used as an index of tortuosity distribution. Curves are presented in Figure 3c and *τ* values are given in Table 4. *τ* was below 1, indicating that the capillaries were almost straight. The median values are 1.22, 1.33, and 1.24 for LV, RV, and S, respectively. Statistical comparisons showed non significant differences between RV, S, and LV. 

### 3.5. Confocal Microscopy

On one heart, both global parameters and architectural parameters of the LV capillary network were calculated from images obtained by confocal microscopy, in order to be compared with the values obtained in the same LV heart from images obtained by light sheet microscopy. A 3D image of the capillary network and the architectural parameter curves obtained from confocal microscopy are given in Figure 4. Data obtained from both imaging methods are given in Table 5.

## 4. Discussion

Our study has established a standardized method of 3D image processing applicable to 3D photonic imaging of the coronary capillary network in mice. This method generates both global and segment-level data allowing 3D quantitative statistical analysis and comparison between samples. Our results provided quantitative characterization of LV, RV, and S capillary networks in normal mice.

### 4.1. 3D Imaging and Image Processing

The iDISCO+ optical clearing method used in this study is known, according to its developers, to induce an 11% volume shrinkage with minor deformation of the brain [11]. Our results showed an estimated heart volume shrinkage around 19%, without significant difference in each dimension, suggesting that, as in the brain, the iDISCO+ method induced a limited and isotropic heart reduction.

One key point for the quantitative analysis of structures such as the coronary network extracted from 3D images is unbiased image segmentation. Due to the poor grayscale resolution power of the eye, a non-visual method was applied for the determination of the threshold values, based on the statistical analysis of the voxel distribution. Because of the overlapping grayscale values of the different objects of interest, i.e., the cardiac tissue and the capillary network, objective identification of the voxel population corresponding to the vascular network was impossible. The segmentation method was hence based on the identification of the cardiac tissue. Indeed, the analysis of non-labelled samples showed that the cardiac tissue corresponded to a homogenous voxel population following a Gaussian distribution. Fluorescence of the cardiac tissue at 639nm has already been reported [18]. For a Gaussian distribution, the mean±1.96SD interval corresponds to 95% of the population. Hence, a threshold value set at mean+1.96SD, as the voxel population corresponding to non-labelled tissue has a normal distribution, ensured the exclusion of 97.5% of non-labelled (non-capillary) tissue. Though the voxel distribution may vary from sample to sample, and despite the overlap of voxel populations, this method ensured that, for each sample, the same proportion of non-capillary cardiac tissue is excluded from the capillary network, allowing unbiased statistical comparison. 

### 4.2. Global and Archhitectural Parameters

Our LSM results showed a number of capillaries close to 350,000 capillaries/mm^3^ of cardiac tissue, with an average length around 15 µm, representing nearly 30% of the cardiac volume, and a total capillary length around 10m/mm^3^ of cardiac tissue. To our knowledge, there are no previously published 3D data on the coronary capillary volume and number density. The coronary capillary number density has been calculated in 2D with histological images, and was estimated at 2500 capillaries/mm² [5,19]. Considering an average capillary length of 15 µm, this corresponds to 170,000 capillaries/mm^3^, a value lower but in the same range of our own values. The total capillary length/mm^3^ of cardiac tissue has already been calculated on rats by 2D measurements and mathematical 3D reconstruction at 3.5 m/mm^3^ [20]. It has also been calculated on mouse cleared brain with values near to 1m/mm^3^ of health cerebral tissue [7]. More recently, total vessel length has been calculated on mouse heart by 3D imaging and found to be 3m/mm^3^ [18]. This value is consistent with our own, though a bit lower. The difference may be explained by differences in image resolution. Indeed, in these studies, voxel size was higher than in our study, and hence may have failed to identify small segments.

The percentage of nodes on the number of segments is a marker of the connectivity pattern of the network since it depends on the relative proportion of the different degrees of furcation (bifurcation, trifurcations, quadfurcation). Removal of artefactual loops may generate artefactual one-to-one junctions between two segments, which may overestimate the number of nodes and segments. However, these possible biases are negligible since the percentage of artefactual loops involved in any type of multifurcation is very low (around 4%).

The fractal dimension is a useful parameter to characterize the complexity of the 3D architecture of a network [21]. Our results showed a fractal dimension close to 2.4, with a relative standard deviation less than 5%, hence with less interindividual variation than the vascular density, whose relative standard deviation was around 35%. Some studies have calculated the fractal dimension of the arterial network in kidney, which was found to be close to 2 [6], and, to our knowledge, it has not yet been calculated for the coronary capillary network. As far as the comparison with the kidney arterial tree makes sense, it indicates that the capillary network is more dense and complex than the arterial one.

Considering the capillary diameters, independently of the species and organ, the value is usually found between 3 and 10 µm, and is estimated around 4 µm in mouse heart [4,14]. These estimations mainly result from 2D studies. Our 3D study aligns with the 2D results observed in the literature as the mean capillary diameter is set around 4 µm. Our results also acknowledge that our methodology based on a lectin injection allows the characterization of the capillary network, as more than 90% of the studied population is composed of vessels with a diameter set between 2 and 10 µm. 

The tortuosity, calculated by different methods, has already been studied on large vessels and on microvessels as a pathological marker [22]. The ratio obtained in our study was characterized by a tortuosity constant inferior to 1, and median values were close to 1, the theoretical value of a straight line, indicating that the capillaries were not tortuous.

Finally, comparison between LV, RV, and S showed that the capillary network is broadly similar in these different parts of the heart, with a lower volume and node density in the S.

### 4.3. Light Sheet Microscopy Versus Confocal Microscopy

Data obtained from CM compared to LSM showed that for scale-free parameters, such as the fractal dimension, the percentage of nodes and the tortuosity constant, the difference between LSM and CM in the same heart were of similar amplitude than the SD of the parameters obtained in LSM. By contrast, the number of segments and nodes, and the total segment length, were much greater in CM, whereas the average length and diameter were lower. These results suggest that CM imaging was able to identify small segments that were not detected by LSM imaging. The difference might be due to the resolution limit of LSM, close to the capillary diameter.

## 5. Conclusions

In conclusion, our study proposes a standardized methodology for 3D image processing of a coronary capillary network that allows the quantification of the 3D capillary architecture and its statistical comparison in different conditions. Thus, it might be used to study pathological coronary capillary networks, and compare them with healthy ones. It is applicable to different imaging techniques, e.g., LSM and CM, and, being based on an open source software, the whole process is easily accessible for a biologist in terms of investment and difficulty level. It also provides a set of parameter values for normal C57BL/6J mice. This study should be completed and confirmed by further studies, taking into account that the scale-dependent parameter values can be impacted by the resolution limit of the 3D imaging technique.

## Figures and Tables

**Figure 1 biology-10-00306-f001:**
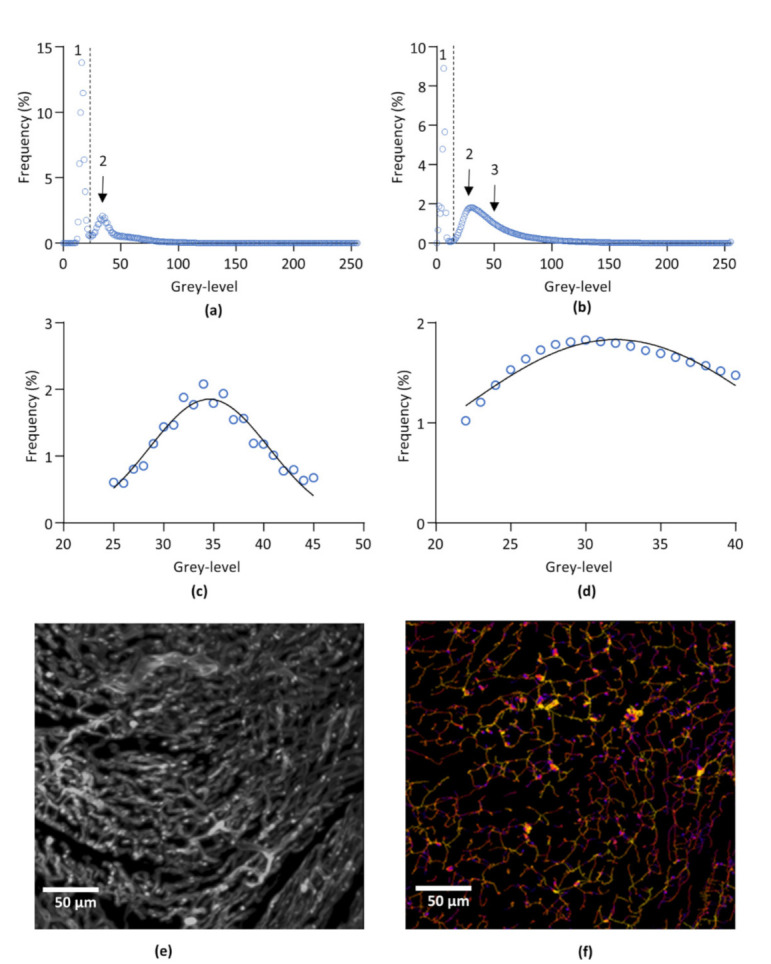
Image processing. (**a**,**b**) Representative frequency distribution of pixel population of non-labelled (negative control) (**a**) and lectin-labelled (**b**) left ventricle after 3D imaging by light sheet microscopy. 1: background. 2: non-labelled cardiac tissue. 3: lectin-labelled capillary network. (**c**), Gaussian fit of the subset of pixels corresponding to the cardiac tissue from curve (**a**). (**d**), Gaussian fit of the subset of pixels corresponding to the cardiac tissue from curve (**b**). (**e**), Representative segmented image of left ventricle capillary network. The image was obtained by overlaying the original capillaries image and the binarized image of the capillaries obtain after segmentation. (**f**), Skeletonization and image mapping of the capillary network of (**e**). Average radii of each segment are encoded in false colors from blue to red. The image was obtained by overlaying the skeleton and the distance map images of the capillary network.

**Figure 2 biology-10-00306-f002:**
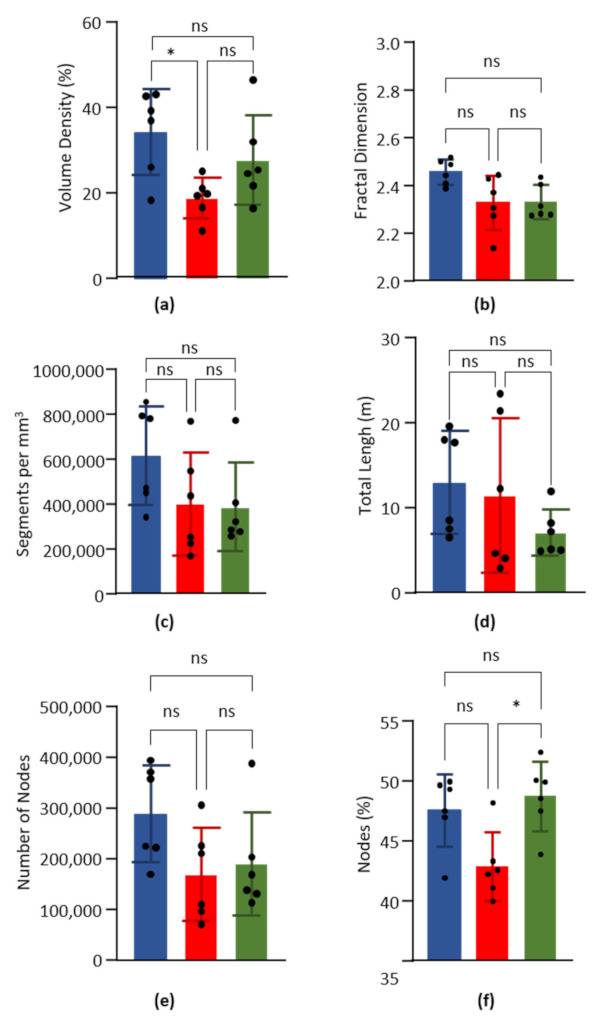
Global parameters. Data obtained from the left ventricle (blue), septum (red) and right ventricle (green) of six hearts. Columns are means and error bars are standard deviations. Black dots are individual values. Data were compared by Kruskal–Wallis test with post-hoc Dunn’s test. ns = non significant. * *p* < 0.05. (**a**). Volume capillary density, in %. (**b**). Fractal dimension. (**c**). Number of capillary segments per mm^3^ of cardiac tissue. (**d**). Total length of capillary segments (m) per mm^3^ of cardiac tissue. (**e**). Number of nodes per mm^3^ of cardiac tissue. (**f**). Percentage of nodes on the number of segments.

**Figure 3 biology-10-00306-f003:**
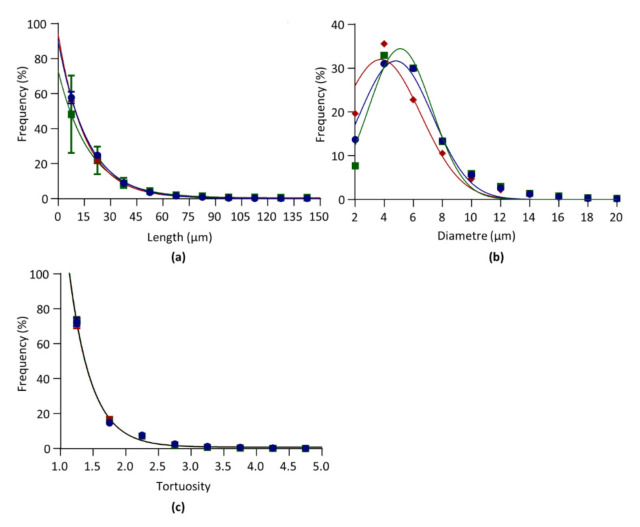
Architectural parameters. Data obtained from the left ventricle (blue), septum (red), and right ventricle (green) of six hearts. (**a**). Relative frequency distribution of segment length. Data were fitted by one-phase exponential decay. (**b**). Relative frequency distribution of segment diameter. Data were fitted by Gaussian equation. (**c**). Relative frequency distribution of segment tortuosity. Data were fitted by one-phase exponential decay. Curves between LV, S, and RV were compared by *F* tests.

**Figure 4 biology-10-00306-f004:**
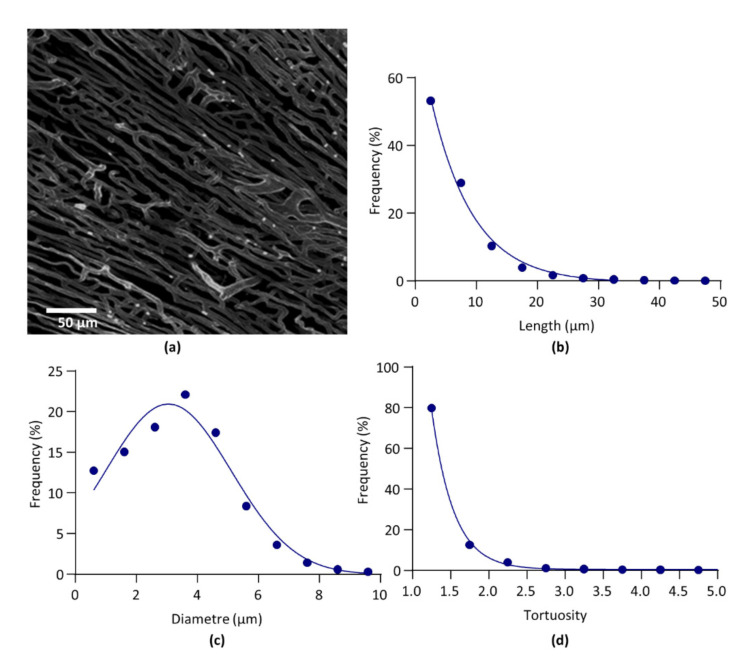
Confocal microscopy. (**a**). Segmented image of left ventricle capillary network. (**b**). Relative frequency distribution of segment length. Data were fitted by one-phase exponential decay. (**c**). Relative frequency distribution of segment diameter. Data were fitted by Gaussian equation. (**d**). Relative frequency distribution of segment tortuosity. Data were fitted by one-phase exponential decay.

**Table 1 biology-10-00306-t001:** Cardiovascular parameters. Systolic (ASP), diastolic (DAP), and mean (MAP) vascular pressure in mmHg and cardiac rhythm in beat per minute (BPM) were measured on six mice.

SAP (Mean ± SD)	DAP (Mean ± SD)	MAP (Mean ± SD)	BPM (Mean ± SD)
123.1 ± 15 mm Hg	89.7 ± 6.9 mm Hg	100.5 ± 9.5 mm Hg	303.6 ± 31

**Table 2 biology-10-00306-t002:** Shrinkage. Length (L), width (W), thickness (T), and ellipsoid volume (V) reduction (in %) after optical clearing were measured on seven mouse hearts.

L (Mean ± SD)	W (Mean ± SD)	T (Mean ± SD)	Volume (Mean ± SD)
6.70 ± 4.2%	10.7 ± 4.2%	2.91 ± 8.2%	19.18 ± 7.7%

**Table 3 biology-10-00306-t003:** Global parameters. Values were calculated from left ventricle (LV), septum (S), and right ventricle (RV) of six mouse hearts.

Parameter	LV	S	RV
Vascular density	34.4 ± 11%	18.9 ± 4.7%	27.8 ± 11%
Fractal dimension	2.46 ± 0.05	2.32 ± 0.11	2.33 ± 0.07
Number of segments/mm^3^ of cardiac tissue	615,784 ± 220,000	399,922 ± 230,000	387,457 ± 200,000
Total length/mm^3^ of cardiac tissue	13.01 ± 6.1 m	11.5 ± 9.2 m	7.11 ± 2.7 m
Number of nodes/mm^3^ of cardiac tissue	289,878 ± 96,000	169,886 ± 92,000	190,392 ± 100,000
Number of nodes/number of segments	47.56 ± 3.0%	42.9 ± 2.9%	48.72 ± 2.9%

**Table 4 biology-10-00306-t004:** Architectural parameters. Values were obtained by non-linear regression of frequency distribution of data from left ventricle (LV), septum (S), and right ventricle (RV) of six mouse hearts (see Figure 3). *λ*: length constant of one-phase exponential decay. μ and *σ*: mean and standard deviation of Gaussian distribution. *τ*: tortuosity constant of one-phase exponential decay.

Parameter	LV	S	RV
Length			
*λ* (mean ± SEM)	16.9 ± 0.6 µm	15.6 ± 0.6 µm	17.9 ± 4.4 µm
Diameter			
μ (mean ± SEM)	4.81 ± 0.24 µm	3.78 ± 0.51 µm	5.12 ± 0.25 µm
*σ* (mean ± SEM)	2.52 ± 0.27 µm	2.75 ± 0.44 µm	2.17 ± 0.26 µm
Tortuosity			
*τ* (mean ± SEM)	0.32 ± 0.02	0.35 ± 0.02	0.35 ± 0.02

**Table 5 biology-10-00306-t005:** Light sheet microscopy versus confocal microscopy. Global and architectural parameters obtained on light sheet microscopy (LSM, voxel size 0.5 × 0.5 × 2µm) and confocal microscopy (CM, voxel size 380 × 380 × 630 nm). The difference between LSM and CM (LSM-CM) values was calculated.

Parameters	LSM	CM	(LSM-CM)
Vascular density	43.3%	31.2%	12.1%
Fractal dimension	2.49	2.53	0.04
Number of segments/mm^3^ of cardiac tissue	794,032	4,297,330	3,503,298
Number of nodes/mm^3^ of cardiac tissue	394,296	1,992,201	1,597,905
Number of nodes/number of segments	49.7%	46.4%	3.3%
Total length/mm^3^ of cardiac tissue	17.7 m	27.1 m	9.4 m
Length			
*λ* (mean ± SEM)	28.9 ± 9.8 µm	6.89 ± 1.0 µm	22.01 µm
Diameter			
μ (mean ± SEM)	4.61 ± 0.68 µm	3.05 ± 0.29 µm	1.56 µm
*σ* (mean ± SEM)	2.25 ± 0.61 µm	2.07 ± 0.26 µm	0.18 µm
Tortuosity			
*τ* (mean ± SEM)	0.31 ± 0.07	0.27 ± 0.02	0.04

## Data Availability

The data presented in this study are available on request from the corresponding author.

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
