# Peer review of "3D Imaging and Quantitative Characterization of Mouse Capillary Coronary Network Architecture"

_biology, 2021, doi:10.3390/biology10040306_

Round 1
Reviewer 1 Report
The presented study is of high quality, especially the style of presentation. It is concise but at the same time presents all the necessary details. One of the weaknesses of this study is statistics, there are some errors in that field. Apart from them I think the manuscript is suitable for publication.
Line 10, there is a not needed comma here.
Line 82, there is a space after “(S)” missing.
Line 182, there is no such thing as “Gaussian law”. There is of course “Gauss law” buit it has nothing to do with this study. Anyway, what you try to express is called “properties of normal distribution”. Please correct.
Line 236, it shouldn’t be “m.mm-3”
Line 252, what about normality tests?
Table 1 and Table 2 and Table 3, the values were rounded improperly, please follow the guides on rounding uncertainties. It is not possible for the uncertainties to have more than 2 significant figures.
Line 401, there is no such thing as “Gaussian regression”.
Line 460, the reported values are valid for C5BL/6J mice, it is not justified to state that those parameters can be compared with other mice.
Reviewer 2 Report
The authors presented a very interesting method for the 3D quantitative characterization of a mouse capillary network. The work has promising application for the analysis of microvasculature under different physio- and pathological conditions. I suggest the consideration of this work for Biology, but I report some major comments.
Major comments
- Authors claimed (see, for instance, Conclusion lines 458-460) that this method can be easily used by biologists, given its simplicity and being based on open-source software. To fully accomplish this aim, authors should highlight which commands and functions should be used at each step of the method.
- Figure 1: are the figure (e) and (f) different steps of the method applied to the same image? If yes, is an overlay possible? Such a picture would help in evaluating the method's ability to skeletonize the network correctly.
- Section 2.7.2 (Filtering). Did the authors try a different type of filtering? If the filtering is aimed at deleting isolated pixels, the function 'Despeckle' can also be used (Process/Noise/Despeckle).
- Section 2.7.3 (Skeletonization). Did authors consider any threshold or filtering during the skeletonization process (e.g. not considering vessels shorter than 2 pixels)?
After skeletonization, authors classified some vessels as an artifact. Please, clarify what a zero-length vessel is and how it can be obtained from skeletonization.
Is such exclusion affecting the network connectivity (e.g. deleting the connection between two vessels)? If yes, how?
Does 'Euclidian distance' refer to the distance from the starting point to the end one? Please, clarify.
Authors should also briefly describe how the radius is computed (based on reference 16) - Section 2.8.1 (Vascular density). The authors chose to compute vascular density as shown in equation (2). That's completely fine. However, when describing the microvasculature in a tissue slab, a different ratio may also be useful. Such ratio involves the microvascular volume and the total volume of tissue. For instance, we can compare a construct's vascularization by considering a sample with the same total volume but with different microvascular volume. The authors performed this kind of comparison in the discussion section (line 409). The equation, in this case, would be Vasc. Density = microvascular volume /(microvascular volume + tissue volume). Please discuss.
- Section 2.8.3 (Fractal dimension). Please briefly describe this parameter and its significance (as done in the discussion).
- Section 2.8.4 (Normalized number of segments). This parameter has been normalized on the tissue volume. Is this the tissue volume mentioned in section 2.8.1? If yes, would the total volume be a more appropriate normalization? In this case, we can compare the grade of vascularization on a sample by considering the number of vessels related to the sample volume. The same consideration applies to section 2.8.6 (Normalized number and percentage of nodes)
- Section 2.8.5 (Normalized number and percentage of nodes). Please briefly describe the significance of the percentage of nodes. Does it represent the complexity of the network? Is this related to the prevalence of bifurcation-like or anastomosis-like junctions? Besides, given the artifacts-cleaning of the network, is there any control of one-to-one junctions? For instance, if a vessel is withdrawn from a bifurcation, the remaining vessels form a one-to-one connection.
- Section 2.8.6 (Segment diameter). Please clarify how the diameter of the vessel is computed. Is that calculated in all the points of the skeleton and then averaged? Was it computed only in one point?
- Section 2.9.1 (Global parameters). The authors showed results by the average and standard deviation. This representation is helpful for normally distributed samples (we know how the distribution is). However, they used a non-parametric test for significance assessment. Please clarify this, either performing a parametric test (if the sample is normally distributed) or showing results with boxplots (median and quartiles). If any other reason for this different treatment of the sample is present, authors should clearly state it in the manuscript.
Similar considerations apply to:
- Results, section 3.2 (the conclusion of isotropic shrinkage is acceptable, but the table not)
- Results figure 2 and table 3
In Tables 4 and 5, results are reported with the standard error of the mean. Please clarify this difference.
- Sections 3.4.1 to 3.4.2 (Global parameters). Please provide brief details on the interpolation technique (software and methods). Report also values of the parameters/variables. Also, explain the function choices.
- Please explain "The difference may be explained by difference in image resolution." At line 420.
- Discussion (lines 438-440). Please provide a reference of tortuosity constant or tortuosity values if contained in reference 21.
- Section 4.3 (Light sheet microscopy versus Confocal microscopy). Even if one sample is not enough to draw any solid statistical conclusion, authors should compare the results with the variability obtained in LSM analysis. Is the sample variability (same area, same conditions) more prominent than the difference resulting from the two techniques? For instance, vascular density for LSM is reported as 34.4± 6 %. The coupled analysis resulted in 43.3% (LSM) and 31.2% (CM), with a difference of 12.1 %, which is bigger than the SD reported. On the other hand, the fractal dimension is 2.45 ± 0.05, and the coupled analysis described 2.49 (LSM) and 2.53 (CM), which is in the range of the SD.
Minor comments:
- Line 236. Please fix the unit m.mm-3 in m mm-3.
- Figure 3b. Please correct 'Diamtre'.
- Line 362. Section 3.4 is not consistent. Correct the numbering.
Reviewer 3 Report
The work of Nabil Nicolas and Etienne Roux "3D imaging and quantitative characterization of mouse capillary coronary network architecture" proposes a standardized methodology for 3D image processing that allows the quantification of the 3D capillary architecture, based on an open source software available to researchers. The work is interesting in itself and the authors are certainly considering “mouse models” of diabetes , hypertension and ischemic heart disease. It would have been interesting to have in this work also some example of note reduction of the coronary capillary network, to validate the methodology, but I do not consider it indispensable.
Round 2
Reviewer 1 Report
The Authors have inclued my comments in the revised version. The manuscript is suitable for publication.
Author Response
We thank the reviewer for considering the manuscript suitable for publication.
Reviewer 2 Report
The authors have greatly addressed the comment of the first report. Therefore, I recommend the publication in Biology.
The last open point is the answer to Reviewer 2 point 8. The authors have explained (in the reply) the possible bias of one-to-one junctions and its negligible effect on the data. Please include few lines in the discussion (manuscript) to highlight this positive result.
Author Response
We thank the reviewer for her/his last comment :
1- The last open point is the answer to Reviewer 2 point 8. The authors have explained (in the reply) the possible bias of one-to-one junctions and their negligible effect on the data. Please include few lines in the discussion (manuscript) to highlight this positive result.
Response: We have added in the material & methods the average percentage of artefactual loops (line 213-214) and added a paragraph in the discussion about the general meaning of the percentage of nodes and a discussion of the possible and limited bias due to the removal of artefactual loops (line 442-448).
Modifications from the previously reviewed version are marked in green in the new revised version of the manuscript.